# Light Penetrating the Seawater Column as the Indicator of Oil Suspension—Monte Carlo Modelling for the Case of the Southern Baltic Sea

**DOI:** 10.3390/s23031175

**Published:** 2023-01-19

**Authors:** Barbara Lednicka, Zbigniew Otremba, Jacek Piskozub

**Affiliations:** 1Department of Physics, Gdynia Maritime University, 81-225 Gdynia, Poland; 2Institute of Oceanology, Polish Academy of Sciences, 81-712 Sopot, Poland

**Keywords:** downwelling radiance, virtual underwater sensor, oil emulsion, underwater monitoring, Monte Carlo simulation

## Abstract

The strong need to control investments related to oil extraction and the growing demand for offshore deep-water exploration are the reasons for looking for tools to make up a global underwater monitoring system. Therefore, the current study analyses the possibility of revealing the existence of oil-in-water emulsions in the water column, based on the modelling of the downwelling radiance detected by a virtual underwater sensor. Based on the Monte Carlo simulation for the large numbers of solar photons in the water, the analyses were carried out for eight wavelengths ranging from 412 to 676 nm using dispersed oil with a concentration of 10 ppm. The optical properties of the seawater were defined as typical for the southern Baltic Sea, while the oil emulsion model was based on the optical properties of crude oil extracted in this area. Based on the above-mentioned assumptions and modelling, a spectral index was obtained, with the most favourable combination of 555/412 nm, whose value is indicative of the presence of an oil emulsion in the water.

## 1. Introduction

Due to the high demand for liquid hydrocarbons and their increased transport by sea, there is a growing need to improve systems for the detection and response to leaks of these substances [1,2]. In sea areas, these substances, whether in the form of crude oil or as products of the refining industry, are transported in the holds of tankers and in the fuel tanks of ships powered by internal combustion engines. In addition, a variety of consumables are in use to ensure the reliable operation of marine equipment [3]. These are engine-lubricating oils derived from crude oil processing and media filling cylinders of reloading devices. Such substances are mixtures of numerous aliphatic and aromatic hydrocarbons. Crude oil also contains various inorganic compounds, and fuels and lubricating oils contain various types of ennobling substances [4]. Oil tanker accidents resulting in extensive spills are relatively rare. However, once an oil tanker disaster occurs, the damage to the marine environment, and often to the offshore livestock industry, is enormous. In this case, the ability to track the spread of such a spill is very important. Hundreds of oil contamination incidents are recorded on the surface of seas and oceans every year. These are generally the effects of unintentional discharges of oil into the sea. Such oil discharges are also penalised as they result from the negligence of ships crews [3]. There are also natural spills of oil substances from the seabed in the offshore oil-producing regions [5]. Failures of shelf mining equipment also happen [6]. The possible sources of oil presence in seawater, related to anthropogenic activities, can also be port canal-origin dredged material dumping [7] and oil installations [8].

If an oil discharge stays on the water surface, it is relatively easy to detect. This is mainly achieved by satellite Synthetic Aperture Radar (SAR) systems, whose signals, after digital processing, provide images of potentially polluted areas [9]. All satellite systems that produce images of the sea in the visible range provide the possibility of tracking the spread of pollution [10]. Airborne scanners sensitive to the reflected ultraviolet radiation of the sea surface provide information on the extent of the oil film on the sea surface. Scanners that record infrared radiation allow the assessment of the thickness of the oil layer [11].

An oil spill located on the surface of the sea can change from a surface form into a deep water one. A factor that favours this process is the high state of the sea surface. In this situation, the oil is dispersed and turns into an oil-in-water emulsion. This process can be artificially supported by spraying an aqueous solution of dispersants over the surface of the sea [12,13,14]. In such a situation, the detection of oil is difficult because the water is opaque to ultraviolet, infrared, microwave and radio radiation. Only the effect of the oil emulsion on the spread of light in the depths of the sea gives a chance to detect oil [15]. The complete characteristics of sunlight spreading in the sea are contained in the spatial, directional and spectral distribution of the radiance defined by Formula (1) [16]. Moreover, the light reflected from the sea surface together with the light coming from the depths of the sea is characterised by the above-water upwelling radiance.
(1)$$L\left(\vec{\Psi }\right)=\frac{dF\left(\vec{\Psi }\right)}{dA\;cos\theta \;d\Omega \;d\lambda }$$
where *dF* is the infinitesimal radiant flux in the $\vec{\Psi }$ direction, *dA* is an infinitesimal area, *dΩ* is an infinitesimal solid angle, *d𝝺* is the infinitesimal range of a wavelength and *𝝷* is the inclination to direction $\vec{\Psi }$. In operational oceanography, radiance is determined for specific wavelengths. A geometric sketch for radiance definition is presented in Figure 1. The part on the left of this figure illustrates the radiance emitted in a specific direction from the sea surface, and the part on the right side illustrates the radiance received from a specific direction by a radiance meter. This is a mathematically exact definition. The solid angle from which the radiance meter receives the radiant flux is finite in value. In oceanological practice, radiance meters are designed to measure at selected wavelengths. In the case of measuring radiance in the sea column, the surface from which the radiant flux originates is abstract—the radiance meter simply picks up the radiant flux from a small solid angle, from a certain direction.

In the case of the research presented in this article, the radiance meter located underwater is directed perpendicularly upwards.

The spatial distribution of radiance in the sea column and the distribution of upwelling radiance above the sea surface depend on the state of the sea surface, the optical properties of seawater and the optical properties of substances in the water in dissolved and suspended forms. An important factor shaping radiance in the sea is the radiance distribution of sunlight falling on the sea surface. Solar radiance is formed in the atmosphere and it is the sum of radiance from the sun and radiance coming from the entire upper hemisphere, i.e., light diffused by the components of the atmosphere [16]. The appearance of an oil film on the sea surface results in an increase in the light reflectance by about two times. Then, the directional distribution of radiance coming out of the water is also modified. Surface oil pollution also results in a modification of the spectral distribution of radiance. If there is dispersed oil in the water column, the light encountering the oil droplets is randomly absorbed and scattered in a random direction. The probability of an oil-induced absorption or scattering of a photon along a certain path in water depends on the wavelength of the light, as well as the size distribution and concentration of the oil droplets. Some of the photons have a chance to reach the seabed and be absorbed or reflected there. Some photons do not end their lives in the depths of the sea and fly out in different directions into the atmosphere, thus forming the above-water distribution of upwelling radiance. Until now, several above-water sensors monitoring oil spills have been studied [17,18,19,20,21,22,23].

In addition to remote methods, the development of underwater sensors could be important for the detection of oil pollution. One of the main reasons for the present studies is an immediate need for the establishment of a practical and easy-to-use system for the detection of dispersed oil. Accordingly, the goal was to simulate the solar downwelling radiance detected by a virtual underwater sensor and examine whether it was possible to detect oil-in-water emulsions floating in the water column.

Earlier studies examined the possibility of detecting oil-in-water emulsions by a sensor located above the sea surface based on the modelling of the directional distribution of the upwelling radiance above the water surface. In the case of a Baltic-type crude oil emulsion dispersed in the water, the most favourable combination of two wavelengths for the determination of the index for the polluted sea area compared to the same index for the oil-free water was identified as 555/412 nm [24]. In the present study, based on the modelling of the downwelling radiance received by a virtual underwater sensor, the most favourable spectral index to detect an oil emulsion in the water column also turned out to be 555/412.

## 2. Materials

The research on the possibility of detecting oil in the sea column using an underwater sensor refers to the southern area of the Baltic Sea (Figure 2). The optical properties of the waters in this area were used to model light propagation in seawater (Table 1 and Table 2), according to Sagan [25] and based on the results of several years of in situ measurements of the absorption and the scattering coefficients for the southern Baltic Sea. The properties of the crude oil extracted in this area and potentially existing as an oil-in-water emulsion with a concentration of 10 ppm were also used in the optical model of the studied sea area (Table 3 and Table 4).

In the study, the polluted water layer is assumed to be 10 m thick (Figure 3).

## 3. Method

The simulation of the light radiative transfer was carried out using the Monte Carlo method, which consists of simulating the fate of a large number of virtual photons. The data necessary for Monte Carlo modelling are the Inherent Optical Parameters (IOPs) of the medium in which the photons travel.

An oil concentration of 10 ppm was assumed. Monte Carlo simulations were used to model the radiance below the sea surface. All stages of the model procedure have been described by Baszanowska et al. [15,26], where the radiance field above the sea surface was modelled. However, in this case, the study was based on the modelling of the downwelling radiance detected by a virtual underwater sensor (Figure 4). In the photon travelling simulation, the water surface is virtually wavy as a result of a wind speed of 5 m/s, based on the Cox and Munk model [27].

The direct photons from the sun and photons scattered in the atmosphere were taken into account and included in the optical model of the sky. It was assumed that the angle of incidence of rays directly from the sun was 30°. Moreover, the lighting conditions were clear sky conditions. The percentage contribution of light coming directly from the sun and light coming from the upper hemisphere of the sky depends on the wavelength (Table 5).

A virtual underwater receiver, acting as a radiance meter, recorded photons coming from a cone with an aperture of 0.14 radians (8.02 degrees).

## 4. Results and Discussion

Radiance meter readings were simulated at depths of 1 m, 2 m, 3 m, 4 m, 5 m, 7 m and 10 m for light wavelengths of 412 nm, 440 nm, 488 nm, 510 nm, 532 nm, 555 nm, 650 nm and 676 nm. Figure 5 shows that the radiance values for all wavelengths rapidly decrease with depth and for a depth of 10 m they are already very low. This is due to the fact that the study covers the southern Baltic Sea waters. These waters are characterised by specific physical conditions, such as the limitation of light penetration under the sea surface and increased shipping and sedimentation of organic and inorganic particulate matter [29]. Despite the fact that the Baltic water transmits light poorly, in the case of slight contamination with an oil-in-water emulsion (10 ppm), the penetration of light is much more difficult.

Figure 6 shows significant differences between the values of radiance for both unpolluted and polluted seawater. The radiance under the sea surface, determined for eight wavelengths, was found to be the highest at 555 nm. However, the greatest differences in the values of this parameter also occur for this wavelength. It is obvious that the downwelling value of the radiation recorded in the sea column depends on the weather conditions, i.e.*,* on the amount of sunlight reaching the water through the sea surface.

Thus, the amount of light reaching the detector cannot be the indicator of contamination. Therefore, the possibility of establishing a correspondence between a specific spectral distribution of radiation and the presence of contamination was examined. For this reason, the spectral index was defined as the ratio of the downwelling radiance for the longer wavelengths to the downwelling radiance for the shorter wavelengths. The study determines the spectral index values for all 28 combinations of wavelengths (Figure 7).

Figure 4 shows that there are considerable differences between the index for seawater polluted with a Baltic-type crude oil emulsion and the index for seawater free of oil. The highest value of the index for seawater with dispersed oil is recorded for the wavelength combination 555/412.

The differences between the values of the spectral index for seawater polluted with a Baltic-type crude oil emulsion and the values of the index for unpolluted seawater were determined (Figure 8).

Figure 5 shows that, based on the modelling of the downwelling radiance detected by a virtual underwater sensor, the most favourable combination to detect an oil emulsion in the water column is 555/412 nm.

In the previous studies [24,30], in which a virtual sensor was positioned above the sea surface, the spectral index values for both oil-free seawater and polluted water were about twenty times lower than when the sensor was located under the sea surface.

The observations presented in this paper, taking into account the conditions of the Southern Baltic both in terms of the optical nature of the waters and the oil extracted in this area, open the possibility of building a model with a wide sphere of universality, considering the nature of the other parts of the ocean.

## 5. Conclusions and Perspectives

The current studies were focused on a radiative transfer simulation to verify the possibility of installing an underwater sensor to monitor the status of marine ecosystems, including their exposure to oil discharges. It turns out that underwater measurements may be much more sensitive than remote sensing measurements. We believe that the development of underwater monitoring systems may be very important for the long-term monitoring of natural and human-induced impacts on seas, rivers and coastal ecosystems. In such applications, optical measurements can be conducted with underwater sensors deployed on various in situ platforms. Therefore, the current study proposes a spectral index of 555/412 for the underwater detection of oil-in-water emulsions in the southern Baltic Sea under cloudless sky conditions. The research could be extended to examine the effectiveness of the underwater sensor method for various oils, weather conditions and sun heights and also take into account the variability of the Inherent Optical Parameters (IOPs) of oil-free water in various sea areas.

## Figures and Tables

**Figure 1 sensors-23-01175-f001:**
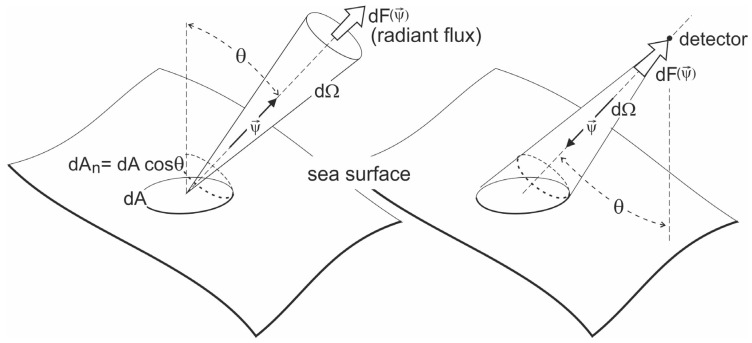
A geometric sketch for radiance definition (after Dera [16]).

**Figure 2 sensors-23-01175-f002:**
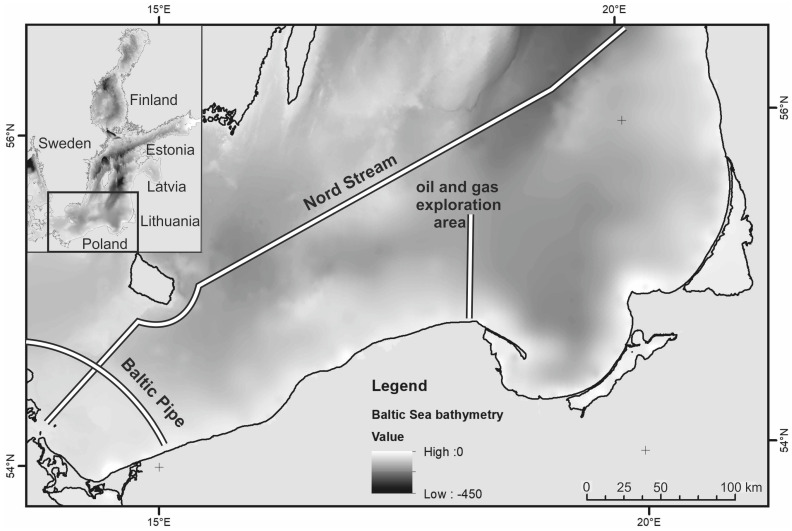
The southern Baltic Sea area for which an optical model of the sea was made.

**Figure 3 sensors-23-01175-f003:**
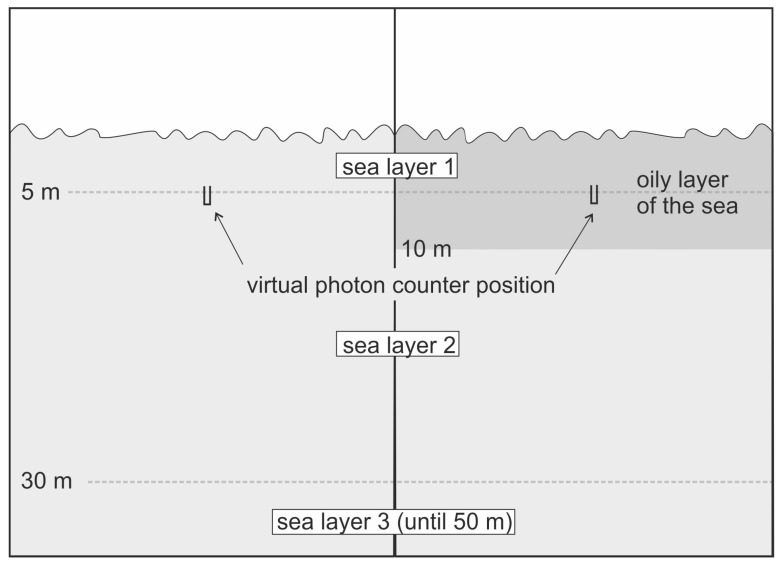
The optical model of the sea basin for eight wavelengths ranging from 412 nm to 676 nm, applied for Monte Carlo simulations of radiative transfer.

**Figure 4 sensors-23-01175-f004:**
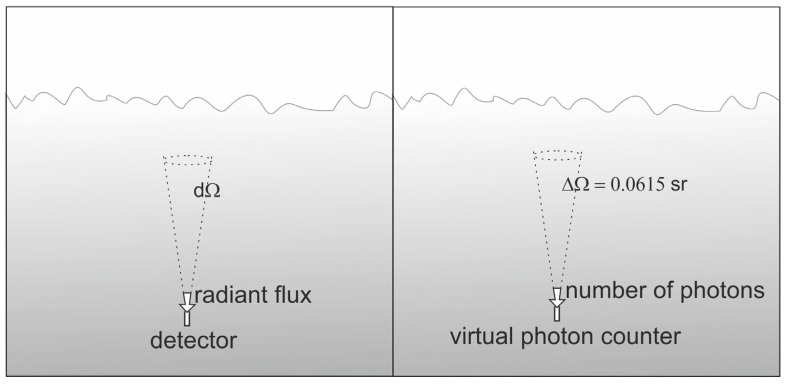
Implemented geometry of radiance measurement in relation to the definition of radiance (**left**) and the geometry of functioning of a solid angle virtual receiver (**right**).

**Figure 5 sensors-23-01175-f005:**
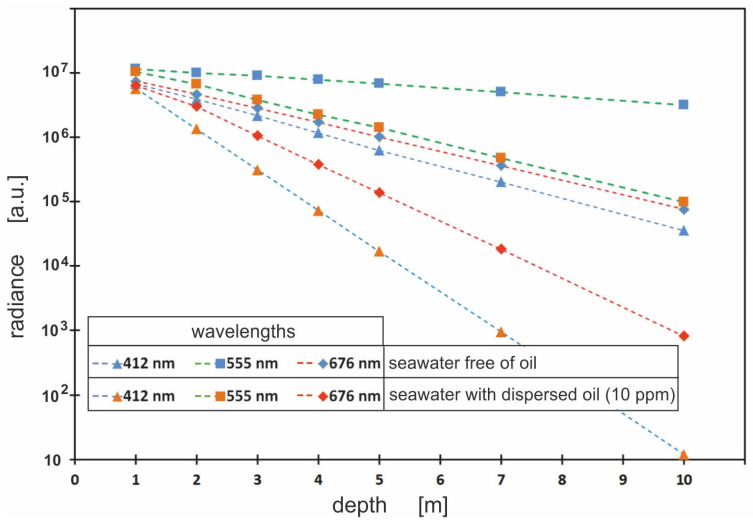
The values of radiance under the sea surface (for chosen wavelengths: 412 nm, 555 nm and 676 nm at seven depths: 1 m, 2 m, 3 m, 4 m, 5 m, 7 m and 10 m), for unpolluted seawater (blue tags) and seawater polluted with a Baltic-type crude oil emulsion (10 ppm) (red tags).

**Figure 6 sensors-23-01175-f006:**
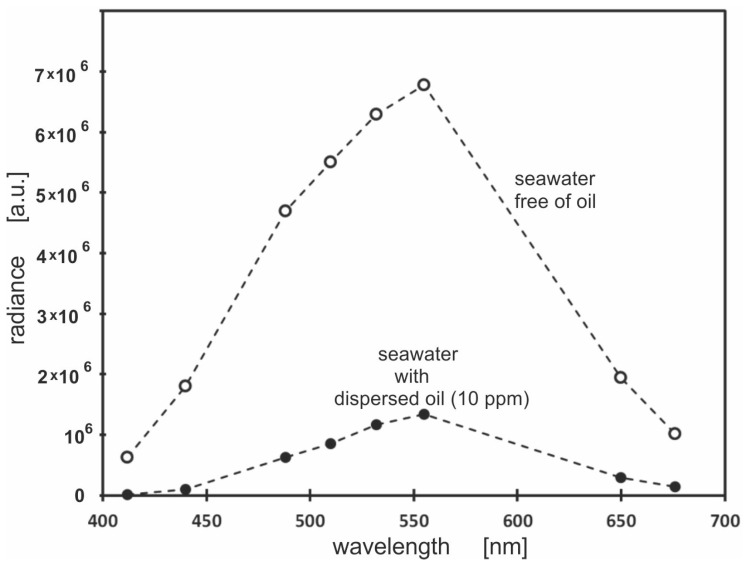
The radiance under the sea surface for eight wavelengths: 412 nm, 440 nm, 488 nm, 510 nm, 532 nm, 555 nm, 650 nm and 676 nm, for unpolluted seawater (white dots) and seawater polluted with an oil-in-water emulsion (10 ppm) (black dots).

**Figure 7 sensors-23-01175-f007:**
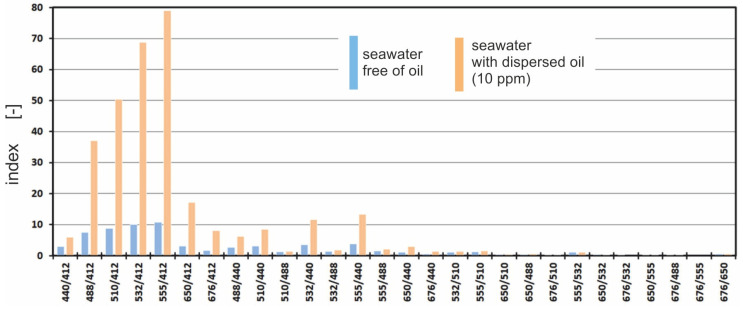
Spectral indices for both seawater free of oil (blue bar) and seawater polluted with an oil-in-water emulsion (orange bar).

**Figure 8 sensors-23-01175-f008:**
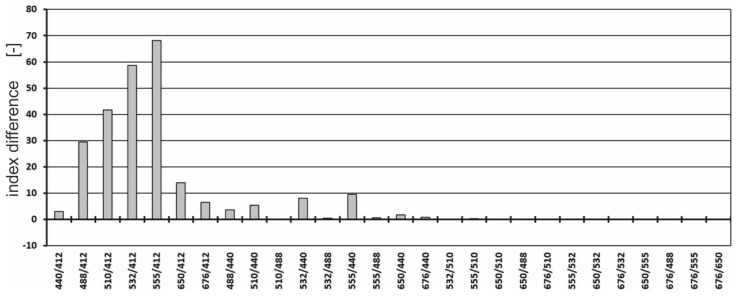
Differences between the values of the spectral index for both seawater polluted with an oil-in-water emulsion and seawater free of oil.

**Table 1 sensors-23-01175-t001:** The absorption coefficient used in the sea environment optical model [25].

Wavelength [nm]	Absorption Coefficient [m^−1^]		
	0–5 m	5–30 m	30–50 m
412	0.596	0.536	0.476
440	0.398	0.348	0.298
488	0.218	0.178	0.148
510	0.188	0.158	0.138
532	0.163	0.143	0.123
555	0.149	0.139	0.119
650	0.391	0.381	0.371
676	0.517	0.497	0.467

**Table 2 sensors-23-01175-t002:** The scattering coefficient used in the sea environment optical model [25].

Wavelength [nm]	Scattering Coefficient [m^−1^]		
	0–5 m	5–30 m	30–50 m
412	0.63	0.39	0.14
440	0.60	0.37	0.13
488	0.60	0.37	0.14
510	0.60	0.37	0.14
532	0.60	0.37	0.14
555	0.59	0.37	0.15
650	0.54	0.34	0.14
676	0.51	0.32	0.14

**Table 3 sensors-23-01175-t003:** The absorption coefficient of an oil-in-water emulsion with a concentration of 10 ppm used in the sea environment optical model [26].

Wavelength [nm]	Absorption Coefficient [m^−1^]
412	0.299
440	0.114
488	0.052
510	0.042
532	0.029
555	0.029
650	0.0125
676	0.0087

**Table 4 sensors-23-01175-t004:** The scattering coefficient of an oil-in-water emulsion with a concentration of 10 ppm used in the sea environment optical model [26].

Wavelength [nm]	Scattering Coefficient [m^−1^]
412	7.81
440	7.97
488	7.98
510	7.95
532	7.91
555	7.87
650	7.60
676	7.48

**Table 5 sensors-23-01175-t005:** The direct solar irradiance contributions used in the sea environment optical model [28].

Wavelength [nm]	Direct Solar Irradiance Percentage
412	64.7
440	66.2
488	68.2
510	68.8
532	69.4
555	69.9
650	71.2
676	71.4

## Data Availability

Not applicable.

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
