# Peer review of "Light Penetrating the Seawater Column as the Indicator of Oil Suspension—Monte Carlo Modelling for the Case of the Southern Baltic Sea"

_sensors, 2023, doi:10.3390/s23031175_

Round 1

Reviewer 1 Report

The author proposed an underwater detection approach of offshore dispersed oil to measure the oil emulsion content. Unlike traditional remote sensing measurements, the underwater scheme could be more sensitive. It is a good idea. However, the Monte Carlo simulation's influence factors are not comprehensive enough to establish an ideal downwelling radiance model. As the author said in the Conclusion part, this research could be extended to examine the effectiveness of the underwater sensor method for various oils, weather conditions, and sun heights and also take into account the variability of the Inherent Optical Parameters (IOPs) of oil-free water across multiple sea areas. Why not add a discussion paragraph to assess or predict the impact of these influence factors.

Author Response

Thank you for your comment and suggestions.

We took the suggestion of yours, for which we thank you, and added the following sentences at the end of the discussion:  “The observations presented in this paper, taking into account the conditions of the Southern Baltic both in terms of the optical nature of the waters and the oil extracted in this area, open the possibility of building a model with a wide sphere of universality, considering the nature of the other parts of the ocean.”

Reviewer 2 Report

The paper is well written, clear, short, and easy to read.  I have two comments though:  The authors pick a concentration of oil at 10ppm.  The optical properties (abs and scattering) for this concentration are very high...absorption at 412 is about half of the background water absorption and scattering is more than 10 times the background water scattering.  It is not surprising that the signal is very evident in the downwelling radiance, but is this concentration of 10ppm reasonable?  These are not subtle changes in the water properties.  So at the least, somewhere (probably introduction?)  some discussion on what relevant concentrations of oil would be, and how 10ppm fits into these.

Second, the measurement they are using, radiance pointing straight up, will be very, very noisy when done in the field.  If the measurements are done at the two wavelengths simultaneously it would perhaps be cancelled out.  This is an unusual measurement configuration, some discussion (or maybe even some data?) about the expected measurement noise expected would be good.

Author Response

Thank you for your comment and suggestions.

Thank you for sharing your thoughts, which will contribute in planning of future research related to the applicability of dispersed oil detection in waters with various optical properties, under different weather conditions, and in relation to possible types of oil and its concentration in water.

As for the choice of oil concentration of 10 ppm, this is lower than the standards for discharge water from ships (15 ppm). Let us add that sea water observed in a glass of water does not differ visually from the same water polluted with an oil emulsion with a concentration of 10 ppm. In the future we will analyze changes in the spectral index value for even lower oil concentrations (than 10 ppm).

It seems to us that starting a discussion on the subject of signal noise at the moment, i.e., in fact, the value of spectral index, would be premature and would obscure the report on the effects of the studies presented in the paper. The effect of a decrease in the sensitivity of the measurement would appear if the spectral shape of the downwelling radiance in oil-free water assumed the same shape as the spectral shape of radiance in oil-polluted water. We will try to investigate in the future how strongly this effect will affect the ability to detect oil.